# Sharing results with participants (and community) in malaria related research: Perspectives and experience from researchers

**Sophie Weston**[1] , **Bipin Adhkari**[2,3] , **Kamala Thriemer**[1] *

**1** Global and Tropical Health Division, Menzies School of Health Research and Charles Darwin University, Darwin, Australia, **2** Mahidol-Oxford Tropical Medicine Research Unit, Faculty of Tropical Medicine, Mahidol University, Bangkok, Thailand, **3** Centre for Tropical Medicine and Global Health, Nuffield Department of Clinical Medicine, University of Oxford, Oxford, United Kingdom

☯ These authors contributed equally to this work.
* kamala.ley-thriemer@menzies.edu.au

**Data Availability Statement:** Data cannot be shared publicly because of the risk that respondents could be identified. Given the context of the interviews even with identifying information

## Abstract

Results-sharing with participants or their communities after the completion of research is an essential element of ethical research. The main objective of this study was to identify and document current practice of trial result dissemination, to explore attitudes among trialists towards result dissemination and to better understand previous experiences and barriers to returning results to participants. This mixed-methods study used a sequential exploratory design with two phases: i) an initial qualitative phase to explore the topic and to inform the quantitative data collection, ii) a quantitative survey. Findings from the survey and interviews were triangulated and presented to a network of clinical malaria researchers for validation. A total of 11 semi-structured interviews (SSIs) were conducted using an interview guide. The quantitative survey had a response rate of 19.9% (42/211). Disseminating results to participants after clinical trials was deemed critical for ethical malaria related research, with 38.1% indicating it as extremely important and 45.2% rating it as mostly important. Most respondents referred to the dissemination of results to policymakers and wider stakeholders as important aspects of research translation. The practice of patient or community engagement was prioritized in the pre-trial period and during the trial for obvious instrumental goals of improving retention, coverage and adherence, but much less priority was given to the post-trial period. The main reason for poor dissemination practice was the notion that the time lag between study participation and the availability of results was too long (42.4%). Other reasons included the assumption that the community was not interested (36.4%), and financial restraints (9.1%). The rich qualitative data revealed detailed accounts of operational, cultural, educational and economic aspects that pose further barriers to results-sharing, including limited knowledge about best practice. Better planning which includes adequate financial resourcing is required for meaningful dissemination of results to study participants. Improved institutional guidance and more stringent requirements by funders could support researchers who are generally interested and willing to complete the process. Best practice methods to conduct such dissemination remains to be explored.

removed, it is relatively easy to identify respondents based on their work they talk about, e.g. clues to where the study occurred and what activities were conducted. However, data are available on reasonable request via emailing to Ethics@menzies.edu.au.

**Funding:** Funding for this work was provided by the Australian Academy of Science, on behalf of the Department of Industry, Innovation and Science through a grant from the Australian Academy of Science Regional Collaborations Program (KT). SW received salary support though a grant from the Bill & Melinda Gates Foundation [OPP1164105/INV-010504]. Under the grant conditions of the Foundation, a Creative Commons Attribution 4.0 Generic License has already been assigned to the Author Accepted Manuscript version that might arise from this submission. The funders had no role in study design, data collection and analysis, decision to publish, or preparation of the manuscript.

**Competing interests:** The authors have declared that no competing interests exist.

## Introduction

Clinical trials are increasingly conducted in low- and middle-income countries (LMICs) because of the high burden of tropical and infectious diseases, like malaria, in such settings [1, 2]. While funders, sponsors and researchers are focused on generating best-practice evidence from clinical trials, much less priority is placed on disseminating results to the study participants [3]. Sharing results with participants or involving participants after data collection, analysis and evidence generation overlaps with principles and values borne by participant (community) engagement for ethical global health research [4]. Communicating trial results to participants is therefore a critical component of adequate community engagement in clinical research. Nonetheless, the bulk of clinical trial literature emphasizes the critical role of engagement at the pre- and during- research stages for reasons inherent in facilitating research, improving recruitment, coverage, and adherence [5, 6]. Participant and community engagement to enhance recruitment and adherence alone however, overlooks the broader ethical goals in health research [7]. Engagement at every stage of the clinical trial including at the dissemination stage is considered to facilitate ethical goals such as respecting participants, improving public trust in health research, building community capacities (promoting health and research literacy) and promoting the bi-directional flow of knowledge [4, 5, 8].

Participating in clinical trials entails complying with the study specific instructions such as offering biological samples, undertaking treatment other than standard treatment offered by the health services, travelling extra miles for follow-up and sacrificing opportunity costs and time [9]. Often, participants are no longer involved after the completion of study specific follow-ups; partly stemming from what is considered the 'endline' in the study protocol [10]. Nonetheless, dissemination of results after the completion of research has long been recognized as an essential component of ethical research, with requirements to provide summary results to study participants in plain language included in the Declaration of Helsinki [11], and current clinical trial regulations in the European Union [12], USA [13], Australia [11] and Canada [14]. Although results-sharing with participants is widely acknowledged to be an ethical practice in research, much less is known about to what extent and how dissemination of results is done to study participants or the community [10, 15–17]. There is paucity of knowledge on methods or strategies around results-sharing with the participants.

The aim of this mixed-methods study was to identify and document current practices of overall trial result dissemination to participants, and to explore the attitudes of researchers towards returning clinical trial results to participants. A further aim of the study was to understand researchers' previous experiences or barriers with returning results to trial participants.

Malaria trials are the main focus of the research team and therefore a starting point for this research, with the ultimate aim to guide and translate the findings into practice, while promoting ethical research.

## Methods

### Overview of study design

This mixed-methods study used a sequential exploratory design [18] with two phases [standard annotation: QUAL-QUANT]: i) an initial qualitative phase to explore the topic and to inform the quantitative data collection, ii) a quantitative survey to provide additional context and extrapolate results from the previous phase across a wider audience. This was followed by an informal presentation of preliminary results followed by discussion amongst members within our clinical trial network [19] to validate the findings.

## Qualitative strand

Details of the qualitative methods utilized in this study are followed using a COREQ guideline (S1 Appendix).

**Sampling frame.** For the initial phase involving qualitative interviews, purposeful sampling and referral within investigators' networks of researchers specialised in the field of malaria clinical trials, across diverse geographical areas, career stage and institutions were used. Respondents in this study were drawn mostly from the pool of expertise in clinical trials in malaria, potential identifiers such as age, sex, years of experience and credentials are deliberately concealed.

**Data collection.** After invitation over email, semi-structured interviews (SSIs) using an interview guide (S2 Appendix) were conducted via Zoom by SW (RN, MPH) who is a research and humanitarian nurse with post graduate degrees in Public Health, proficient in qualitative research methodologies. SSIs, data analysis and synthesis were supervised by KT (MD, PhD) with extensive experience in quantitative and qualitative methods, malaria related clinical trials and BA (MD, PhD) who has expertise in social science related research including community engagement. The structure of the interview guide was flexible and iterative, but discussion was broadly steered towards the following three themes allowing exploration of: i) general opinions of sharing results with research participants in clinical trials, ii) experiences with specific studies in which they did/did not share results to participants, iii) perceptions of specific challenges in sharing results with participants.

In addition to introductory emails explaining the objective of the research and requesting their participation, respondents were informed about the study, its rationale, risks and benefits using a standard participant information sheet and consent form before the commencement of interview. Participants were invited to speak freely regarding their experiences, and for this reason anonymity of the participants has been closely observed. All interviews were recorded using the Zoom recording function. Most of the respondents did not have prior familiarity with the interviewer (SW). Respondents were acquainted through invitation emails and informal interactions before the interviews. For example, a few minutes were invested in building rapport before commencing each interview. Each interview lasted between 30 minutes to 1 hour. In one instance, part of the transcript (and its interpretation) was chosen to illustrate an example of the result-sharing practice in malaria clinical trials. Potential identifiers were avoided based on the feedback and request for anonymity from the respondents. Audio recordings of all the interviews were initially transcribed using Otter.ai and were manually cross-checked with the audio recordings.

**Data analysis.** Thematic analysis of the transcripts was conducted to inform the priority areas of further exploration and the survey questionnaire. All transcripts were collated in QSR NVivo 12 for line-by-line coding and were coded by BA and validated with SW and KT. Data were categorised based on the broad codes derived from the SSI guide and were subsequently supplemented by additional codes to incorporate the emerging themes from the transcripts. Themes, together with the supporting data, and their relevance were discussed among the authors (SW, KT and BA). Any disagreements related to data coding and supporting evidence were resolved by discussion. Final themes and the supporting data were further refined aligning with the research question and the quantitative survey findings.

## Quantitative strand

**Sampling frame.** Two major clinical trials registers were used to identify potential participants: i) ClinicalTrials.gov and ii) ISRCTN registry. The search was limited to malaria trials conducted within the last 15 years, including both individual and cluster randomized trials, completed and ongoing, including adults and children and all geographic areas and limited to

phase three and four trials. Duplicates that were detected in both databases were removed and contact details (email address) for listed principal investigators were searched for manually.

## Data collection

The survey instrument was developed based on the preliminary analysis of qualitative data (S3 Appendix). A total of 16 questions were included focusing on i) general opinion on utility of disseminating results of research to participants in clinical trials, ii) experiences from previous studies in which they did/did not return results to participants, iii) perceptions of specific challenges or barriers in returning results to participants. The survey instrument was designed in Qualtrics using predefined skip patterns and logic to direct respondents to the next relevant question depending on previous answers.

Automated email distributions were sent out including a link to the survey instruments. To assist with completion rates, three subsequent reminder emails with attached survey link were sent out through the same electronic channel, at intervals of one week, two weeks and eight weeks post initial email.

**Data analysis.** Basic descriptive statistics and text responses for each question including the simple charts that are automatically generated by Qualtrics were used. In addition, illustrative free text from the quantitative survey were integrated into the qualitative findings.

## Participant webinar engagement and validity workshop

Findings from the SSIs and the survey were summarized into presentation slides and presented to an existing malaria clinical trials network [19]. The webinar was utilized as a forum for discussion (validity workshop) to assess whether the findings resonated with the participants. A total of 20 attendees participated in the webinar and consisted of a diverse group of skills and capacities that included clinician researchers, scientists, data managers, social scientists, and study coordinators from a mix of low- and middle-income countries to high income countries. The webinar attendees were from South Asia, Southeast Asia, Sub-Saharan Africa, and Australia. Some of the attendees of the webinar were also respondents of this study. The discussion findings largely resonated with the results of the study and added some new feedback. Implications of the discussion findings have been integrated into the discussion section. Based on the thematic analysis and findings from the webinar, data were deemed to be saturated as no new themes were further identified [20].

## Ethics approval and consent to participate

For the qualitative interviews, the participant information sheet was provided via invitation email. Prior to commencing the survey, written pre-approved informed consent and verbal informed consent to continue and to record the interview were obtained. The quantitative survey included the participant information sheet and an informed consent form required to be completed before commencing the survey. All research was done in in accordance with the Declaration of Helsinki and the study protocol was approved by the Human Research Ethics Committee of the Northern Territory Department of Health and Menzies School of Health Research (HREC #20–3801).

## Results

### Participants

For the initial qualitative phase, a total of 13 potential respondents were emailed requests to participate in the study and 11 confirmed their participation. No reasons for non-response

were identified. These 11 interviews were conducted between November 2020 and February 2021. All respondents had previously been principal investigators on malaria clinical trial/s, with four predominantly working in Asia, six predominantly in Africa and one in the Americas.

The sampling frame for the survey included 293 participants to whom the survey was emailed in December 2021. A total of 34 (11.6%) emails failed, 41 (14.0%) bounced and 7 (2.4%) were duplicates that were missed previously. No reasons for non-participation were recorded. A total of 13 respondents participated within the first two weeks, after that, three rounds of reminders were sent (December 2021, January 2022 and February 2022), resulting in a total response rate of 19.9% (42/211).

## Rationale of result dissemination

The majority of respondents (88.1%; 37/42) felt that results should definitely be communicated regardless of the outcome of the trial. Dissemination to trial participants itself was felt to be extremely important by 38.1% (16/42), while 45.2% (19/42) felt it as mostly important and 16.7% (7/42) thought it only has little importance.

Almost all respondents of the SSIs agreed in principle that results of the research should be shared with the participants as part of good research practice and laid out distinct reasons for doing so (Fig 1). One of the reasons for sharing results with the participants after the end of the trial was the rationale that since the participants contributed to the rigors of the trial, they were entitled to know how their biological material had been interpreted or yielded an outcome.

*I think [results-sharing] should be done. Yes. Especially. . . they're giving us samples from their body. [. . .] So, I think it would be very good, if we could give back all the information to the*

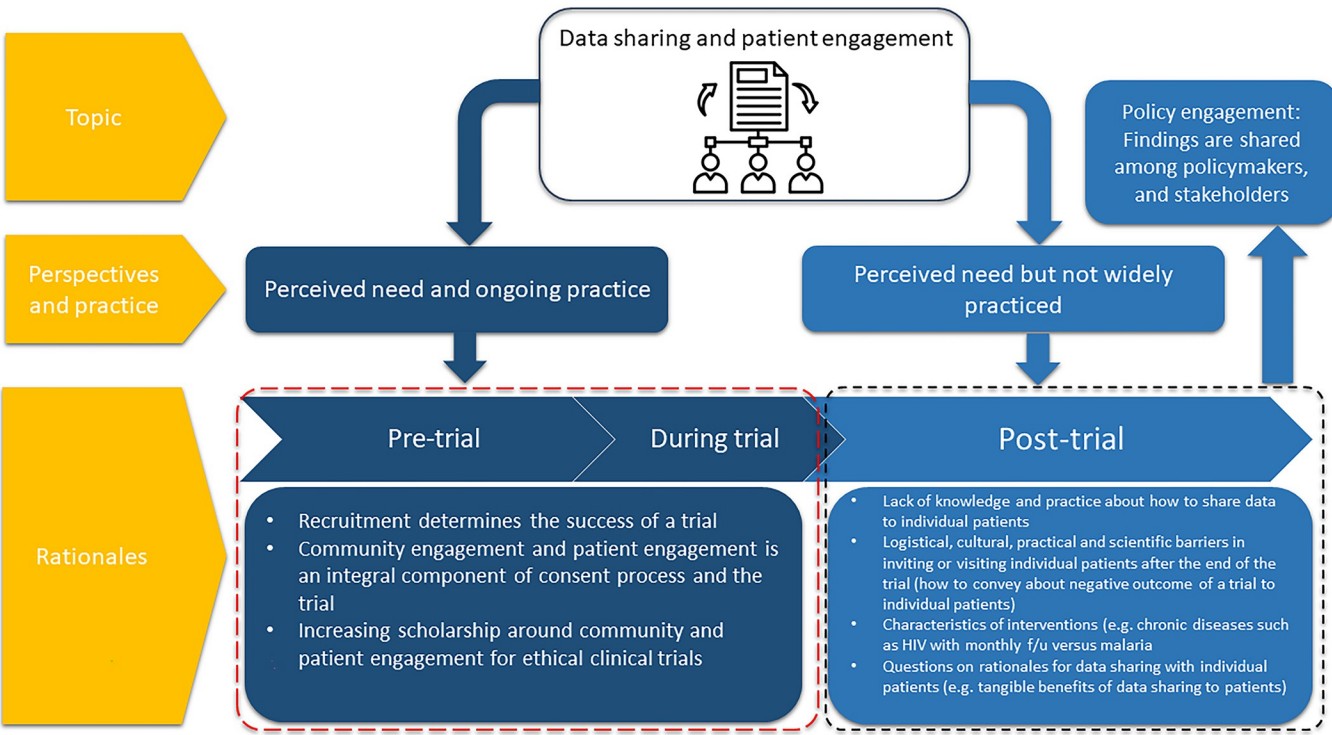

**Fig 1. Schematic diagram showing overview of rationale for community and patient engagement across the trial life-cycle.**

*participants itself . . . I think they deserve this because they participated in our study*. SSI-1 (Asia Region)

At least two respondents, from both Africa and Asia regions, referred to some of the pitfalls of conducting research, where due to resourcing, researchers must prioritize and spend most of their available time on pre-trial and during trial engagement with the participants and cease engagement after the collection of the final samples or the last study visit. Such practice was considered to put the reputation of researchers and institutions at risk, eroding trust and relationships with the participants and their communities. Respondents also rationalized the sharing of results as a moral obligation and explained it as an ethical component of research. Timely post-trial results-sharing was seen as maintaining connection to the community and participants by promoting relationships and trust towards the researchers and the institution and were perceived to be critical for the uptake and acceptance of health information supplied to the community.

*And I think that's why feedback represents a very important way of building confidence in research participants, and, you know, in the long term, because I think that's one of the things that . . . really helped me to understand that research partnerships are a long-term process, it takes a long time to grow, but very short time to, you know, to destroy. Giving feedback, even if the results are not say, statistically significant, I think . . . it's important that it's part of the building trust relationship with the community that we provide that feedback in time*. SSI-2 (Africa Region)

Feedback was also seen as beneficial to counteract misinformation, rumors and minimize any perceived sense of exploitation of the participants. Sharing of results with the participants was also described as being responsive to the participants by being accountable and transparent about the research activities, which leads to strengthened relationships and trust within the community.

*I guess that's part of making sure any message you're ever disseminating is incredibly clear as to what we're doing, why we're doing it, the benefits for the community, the benefits for the country, but also the fact that we are using samples appropriately*. SSI-9 (Asia Region)

*It may also help in increasing the level of relationship you have with the community you're working with, just to say you're not a guinea pig, you know*. SSI-3 (Africa and European region)

Some respondent questioned the rationale of sharing results with community members, and specifically questioned how sharing of results would yield tangible impacts such as policy change or incorporation of alternative treatment regimens in their community. For one respondent, the relevance of sharing results with participants, years after the trial ended, was raised as a point of disagreement. Other respondents simply valued the ethical aspects of results-sharing including how it could promote the overall knowledge of research and health, contributing to a community's understanding of research and therefore future acceptability of health interventions and research.

Echoing the qualitative interviews, most survey respondents who reported to have shared trial results in their practice felt it was useful (87.5%, 21/24). The rationale for sharing results with research participants was explained as participants' rights to know their contribution in

research, that the process engendered respect to the participants, fostered transparency about research, promoted research literacy, and built relationships and trust for ongoing and future research.

## Current practice

Although posttrial results dissemination was undoubtedly perceived to be important, the dissemination of results to participants after the trial was not widely practiced with only 9 out of 42 respondents (21.4%) indicating that they had disseminated results of their last 3 or 4 trials and 16 respondents (38.1%) reported that they had disseminated results for some but not all of their trials.

When asked about the sharing of results to different stakeholders, most respondents disseminated to funders (78.6%, 33/42), their peers (through publication or conference) (88.1%, 33/42), to a Ministry of Health (71.4%, 30/42) and/or to local health authorities (57.1%, 24/42). However, only 35.7% (15/42) reported disseminating results to community leaders, and 23.8% (10/42) to trial participants (Table 1).

When asked about post-trial results-sharing, some respondents in the qualitative strand of this work gave examples of pre-trial and during trial engagement with communities and patients, somewhat conflating the distinct stages. Nevertheless, sharing of post-trial findings with policymakers, authorities and relevant stakeholders in the decision-making capacity was rigorously conducted, while sharing results with participants was often aspirational (Fig 1).

This was further echoed by our clinical trial network meeting members.

*There is not anything formal [in relation to methods of disseminating of results], yet we have been discussing to start doing that. Or we have current trials going on that we are discussing with people at our institution to do that, but we haven't so far done that systematically and we haven't come up with a way of doing that yet.* SSI-5 (Americas Region)

There were many motivations to include aspects of post-trial data sharing into prospective and upcoming research plans. A total of 60.0% (27/45) reported in the survey that they are planning to implement research results-sharing in their upcoming trials. A number of researchers explicitly shared their prior practice of results-sharing, although it was noted that they were not able to, or chose not to, evaluate how much they were able to reach to their participants and the impact of the experience on the participants.

**Table 1. Result dissemination to different stakeholders.**

| Type of Stakeholder | N (%) |
|---|---|
| Peer reviewed publication/ conference | 37 (88.1%) |
| Funders | 33 (78.6%) |
| Ethics Board | 22 (52.4%) |
| Ministry of health | 30 (71.4%) |
| Local health Authority | 24 (57.1%) |
| Study Clinicians | 30 (71.4%) |
| Other study staff | 23 (54.8%) |
| Community leaders | 15 (35.7%) |
| Media | 12 (28.6%) |
| Trials participants | 10 (23.8%) |
| Other | 3 (7.1%) |

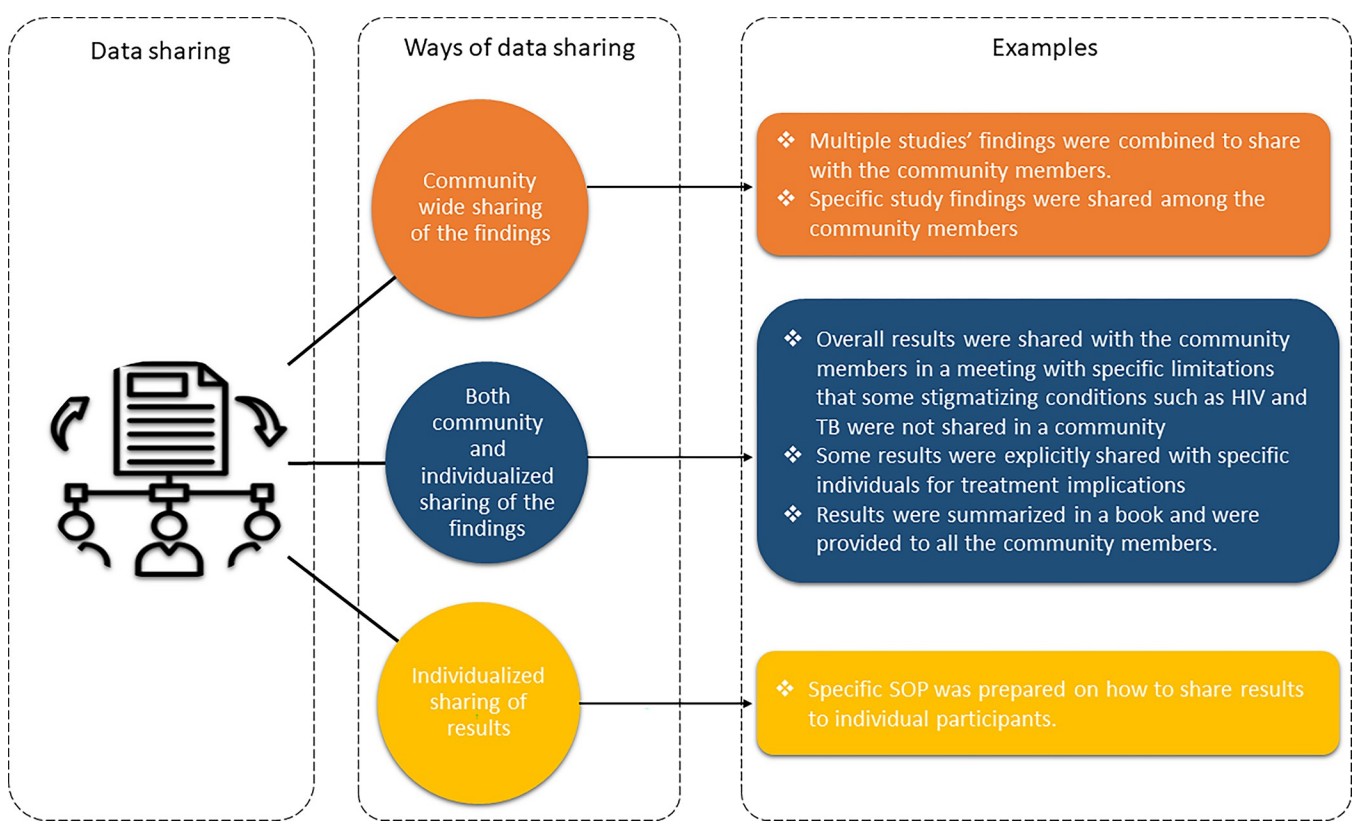

**Fig 2. Potential ways of results-sharing with the participants/community.**

### Different approaches

Although most of the researchers acknowledged that they did not have standardized post-trial results-sharing practice, some respondents were able to report methods used (Fig 2). Among those who shared their experience of results dissemination at the end of the study, the dissemination was not clearly defined as a distinct event but were rather integrated with other community and authority engagement activities. A few noteworthy instances included the implementation of dedicated post-trial strategies for sharing results, which involved the development and utilization of standard operating procedures (SOPs). One participant in the Qualtrics survey component shared an institutional standard operating procedure (SOP).

*So the SOP is necessary. Basically, it describes the process, and then there's a template. So the paragraph. . .can be adapted for each particular trial. And then that has to be approved by the principal investigator. And then there's a log for the trial staff to use to record their attempts to get in contact with patients. I mean, this is a malaria trials on a border area with mobile population, a lot of undocumented migrants. So it's quite a difficult situation there anyway. I mean, obviously, the SOP is generic for use in any trial.* SSI-6 (Africa region)

*Yes, we do that [dissemination results to the community/participants]. In fact, we have a standard operating procedure, which was developed specifically to address that issue. And our ethics review committee does require that we disseminate results back to the community. Our SOP specifies that we have to start first with the study staff, and then from the study staff, then we have to inform the Ministry of Health at the National and county level, [. . .]. And the*

*reason we do that before we go to the community. . .including community advisory board, and the study participants, the way we do that stepwise is that we don't want to go out into the community before the health authorities know about that research, because if the information reaches them second-hand, they will start asking 'who gave you the permission to talk to our community without involving us' because if anything negative happens, then we'll be the first one they will come to. So, we inform [the Ministry of Health] first and also the results, whether they're negative or positive, and then we go to the study participants and then the community.* SSI-8 (Africa region)

Based on the survey results, it was found that a general community meeting emerged as the most frequently employed method for dissemination (57.1%, 20/35), with very few reports of meetings exclusively for participants (11.4%, 4/35), social media campaigns (8.7%, 3/35), generic emails to participants (5.7%, 2/35) and individual phone calls (2.9%, 1/35). Those who reported employing other methods listed focus groups, meetings with community leaders, other public gatherings and informational print-outs that were distributed to community members. Consistent with the qualitative findings, researchers who were able to provide insights into their post-trial results-sharing practices reported a preference for conducting community meetings rather than sharing findings on an individual level. Additionally, some respondents explained that the findings from multiple studies were combined while presenting to community members. Adequate time, typically two to three weeks, and appropriate funding were allocated for such activities. Furthermore, researchers in certain cases reported selectively sharing results with individual participants, particularly in sensitive contexts, for example with conditions such as HIV where confidentiality and stigma are major concerns.

*We had a large meeting, which took us about six months to organize, because we wanted to bring in WHO as well as the National malaria control program. So finally, we did this whole kind of a big meeting, workshop, whatever you want to call it, and, brought in all the participants, all the other people who are interested {. . .} and had a meeting for it, and then wrote that up and published it, which I think was kind of the most concerted effort of giving feedback to the population where the study took place.* SSI-4 (Africa and Asia Region)

Researchers also offered examples of potentially expanding the use of social media (e.g., Facebook, Whatsapp groups) to disseminate the trial results with participants, but also acknowledged that these had limited utility in inadequate network coverage, or very remote settings. While application and uptake of such methods was deemed to be imminent, their full scale roll out is yet to begin. Another potential method was preparing audio-visual messages and sharing through social media.

*So ideas we have is through using social media to disseminate the results and what we've been doing now for some of these trials. . . [is to] have a Facebook group. . .that's some of the strategies we have been discussing and haven't done yet we were planning to start this year. But with COVID. . .we are trying to adapt [and] that's when the issue of social media or other things have appeared. But for rural settings, it's not too much applicable.* SSI-5 (Americas region)

Adapting resources to the context to optimize the results dissemination was an important aspect raised. This included using lay summaries, pictorial representations and interpreting scientific data to be translated by local healthcare workers.

*All of these documents are written by my colleagues, because they can better reflect the language that is required in order to communicate the findings. So usually, one of my study doctors who is a national, she'll prepare the statements. They're very much a lay version. And so, it will basically be 'this was the main result'. . .And presenting scientific data obviously, requires sometimes a little bit of interpretation. But I guess to be honest, it depends on how its prepared. If you had a very simple written document, maybe with pictures or something, you probably could communicate that information quite effectively, without, you know, complicating or causing confusion.* SSI-9 (Asia Region)

In terms of practicalities of results-sharing, certain respondents shared how they integrated the dissemination of results into their research and effectively demonstrated both the inherent challenges presented in this study but also potential strategies to mitigate them (Table 2).

## Barriers to results dissemination

There was a large overlap observed between the barriers preventing the targeting of participants and the challenges encountered when attempting to do so. Most of these barriers were

**Table 2. Process illustration of integrating results-sharing with the participants in research practice (Author-edited narrative based on the conversation with a researcher from Africa Region).**

| | |
|---|---|
| **Practice** | We mix multiple clinical trial results and disseminate to the participants together. For this, we have decided to allocate some time ranging from two weeks to three months that entails traveling to participant villages where we have conducted our clinical trials or other community based (non-clinical trial) studies. We organize a community meeting in which we invite all stakeholders that include all community members, and all formal and informal authority members. |
| **Challenges and tailored strategies** | But sharing scientific findings with the participants is not easy, particularly because of the complexity of the science and potential of incomplete or misunderstandings. To ensure that community members understand the study findings, we delegated a local community member familiar with the local language. We made a summary of the results and was shared with the translator to prepare it for translation with the community. But science translating is difficult, and we need a communicator who has some level of understanding of the science and research. We also used role play to articulate the results and their impact to the community, mostly to make it more fun and engaging. It was really interesting and entertaining to them as well. |
| **Rationale for sharing results with the community** | Why should we provide feedback to the community? Because research, you know its not just come, take and go. Yeah, because they are taking part to the study, they need to know what they have contributed to. The more they take part, the more they understand the value of their contribution, even though initially community members did not understand what research is and why they should participate. But when we go to the community and share the findings of the research in which they participated, they will understand more the reasons why, so that you build a strong base in the community for future participation. So, you have to put yourself into their shoes, consider yourself as a community member and go to discuss when you have findings. |
| **Reflections on initial adoption of results-sharing in research** | I can confess that initially since we did not share the results with the participants, we noticed that people want something, they were kind of reluctant to talk to you. You could know that something is not right. You know they want to know 'what's going on'. So, this is one of the main reasons why we considered this, the importance of feeding back to the community. We realized and from now on, we have started 'result-sharing' in our practice. And as soon as we have findings, rather than going only for single research, we wait for five or six, research and we share it together, this is also more cost effective for us. |

rooted in the logistical, cultural, practical, and scientific characteristics of the study and the country where it was conducted. For instance, lack of planning from the beginning, together with a long time lag between trial completion and data analysis emerged from the qualitative interviews as the main barriers to results dissemination. This sentiment was echoed in the survey results, with a large proportion (14/33, 42.4%) identifying time lag as the primary reason for not targeting participants in their dissemination activities. Other reasons included the assumption that the community is not interested (12/33, 36.4%), and financial restraints (3/33, 9.1%).

Organizing results-sharing with individual patients or the community was considered a huge undertaking and seen as requiring resources for engagement comparable to pre-trial and during-trial investments.

*And then . . . trying to get that information back to the community from where we recruited. This is always the most challenging one, because obviously, quite often, quite a decent period of time has passed since the actual trial was run to when we have official results that we can disseminate back.* SSI-9 (Asia Region)

Researchers recognized the barriers inherent in their work-cycle and the level of commitment required from undertaking. In terms of researchers' priorities, their cycle of work, which involves conducting research, publishing it and writing subsequent grants, often leaves little time to invest in post-trial results-sharing. Simply lacking dissemination planning in the study protocol or application for grants/funding was also attributed as a major barrier, as resources (time and money) are required for organising community meetings, finding participants, travelling to the sites, and engaging community leaders. Lack of adequate funding for these activities was identified by 13/25 survey respondents (52.2%). Not investing time (or not having time available) for post-trial results dissemination was recognised as a significant contributing barrier to sharing results at the completion of research.

*I think, for most times. . .towards the end of the research. . . the researcher is trying to beat some deadline of funding or trying to finish up, you know, some analysis or publication that they often. . .overlook this point. Most times, I don't think it's a funding issue. I think it's really having the conscious time commitment to sort of put together a with a feedback response to the community. I think that's probably where this needs to be focused.* SSI-2 (Africa Region)

Malaria trials are often conducted in remote locations where populations in those areas can be highly mobile. Finding the participants after the completion of the trial (2–3 years later) was thought to be difficult and was raised by 12/25 (48.0%) of survey respondents as a challenge. Participants may have already moved to new places, changed their phone numbers and it is possible they may not consider the results to be of relevance, or interest.

*That's something that we first see specially for the malaria studies because in some of the areas we work, people move a lot. So sometimes people [in] a rural settlement or in a mining setting . . . go to the big city [or] into the mining areas or vice versa. And then they change their phone numbers. So they are not present anymore. And if it's a very long study [the time] from the first participants to the results can take two to three years or eventually more to publication and it may be difficult to find them again, and they may not be very interested directly in that result, if they have moved to an area where malaria is not that relevant anymore.* SSI-5 (Americas Region)

Researchers also reflected based on their past research where they had a mixed experience of sharing results, specifically because of the complexity of sharing scientific results. Challenges identified were a perceived lack of community interest (6/25; 24.0%) and more importantly low literacy and difficulties explaining complex results (20/25, 80.0%). For instance, negative, or in-significant outcomes from the trials are not easy to communicate to the participants. Even mixed results are complex to explain to the participants. In such scenarios, it was a concern to researchers that sharing of results (negative or complex) may fuel resentment from participants towards the study, jeopardizing future research.

*How do you explain . . . to a population . . . well, it was kind of a mixed success, do you say it was no success? It would be really easy to go back to a community and say, 'look full success, done really well'. Okay. But to go back and say it's what you call a conditional success, or you don't want to admit that it is a failure, it becomes much more difficult.* SSI-4 (Africa and European Region)

### Formal guidance by institutions

Although research institutions recommended and encouraged researchers to engage public and community with their research outcomes, there were little accounts of explicit institutional mandates for researchers to share their post-trial results with the participants. This was in line with the survey, where only 22.5% (9/40) reported to have institutional guidance for result dissemination. In addition, some of the survey respondents mentioned the lack of institutional interest as a major barrier (2/25, 8.0%).

*You do the disseminations, but there is no requirement [for dissemination], you can publish it. It may be what is required from my work and the report [to] the regulatory agencies, but these kind of workshops? It is encouraged but it is not a requirement to do.* SSI-7 (Africa region)

Increasingly, funding applications including institutional review committees are requiring the submission of plans for result dissemination to participants, community, public and scientific networks.

### Discussion

Sharing results after the completion of malaria research with the participants and community members was acknowledged to be an essential aspect of ethical research. Specifically, participants as providers of data were deemed to have rights to know the final outcome of the research and its individual and broader implications. Sharing of results was also considered to strengthen the transparency, accountability and trust between participants and the researchers. Despite research ethics requirements, current practice of results-sharing was poor and has been shown previously in academic outputs [16]. Despite some study participants sharing their current approaches, barriers to dissemination emerged at various levels including logistical challenges, funding limitations and poor formal guidance by institutions.

Respondents in this study inevitably interconnected results dissemination with how they have been engaging and sharing their data with the policymakers and wider stakeholders (e.g., WHO, scientists, national malaria programs, and funders) but not the participants or community members where the research was conducted. This also reflects the wider beliefs and practice of researchers in global health to selectively prioritize their results dissemination to the

relevant stakeholders (e.g., scientists, policymakers, and funders) where the research can be translated into policy, referred to in future work and funded [17, 21, 22]. Maintaining this priority often serves researchers to promote their career related achievements [17]. Although disseminating results to participants may not offer tangible impacts such as policy change or provision of alternative treatment to the community members, it serves to address ethical research principles of showing respect to the participants, by ensuring transparency and building trust [23] and thereby leading to increased uptake of health innovations.

A large survey of clinical trial investigators working in 71 different countries found that 33% of investigators did not plan to disseminate their results, while only 13% planned to do so [17]. When dissemination did occur, almost half shared academic reports as opposed to including lay summaries in simple language. The study concluded that dissemination of study results to participants was not yet incorporated as an essential component of the research process.

Result- sharing can also be conflated with the interim individual results-sharing (feedback) during the trial rather than exclusively sharing results after the completion of the trial. Although still in formative stages, development of guidelines and standard operating procedures can add evidence and support to post-trial engagement and results-sharing [16]. While social media platforms are relevant for post-trial results dissemination, their access, and skills to use them, particularly in rural communities where research are conducted, are major impediments [24]. There are also emerging questions around modality of sharing results, that is individualized study results-sharing versus sharing in groups or communities. Both modalities have their own merits. For example sharing individual results due to issues of confidentiality in HIV research and challenges, including the practicalities of convening participants after the trial and bears important implications when planning results dissemination.

All respondents echoed the need to strengthen the results-sharing with participants or community members in research and offered a variety of rationales. Rather than participants being considered as solely sources of data for research, results-sharing was viewed as an ethical engagement practice, particularly as it can strengthen the transparency between researchers and participants [15]. Amidst the negative historical (past colonialism and exploitation) [25, 26] and current global health research practices such as multi-country HIV trials [27, 28] and recent Ebola trials [29–32] where participants and community members were inadequately engaged, results-sharing after the completion of research adds to a much-needed transparency critical to build relationship and trust between researchers and participants [4, 15, 16]. Declining trust in scientific integrity and emergence of science denialism globally further urges the scientific community to invest in overall community/participant engagement [33].

Although results-sharing with the participants/communities shares principles and values borne by community engagement for ethical global health research, comparatively, much less effort has been placed on engaging participants after the research (post-trial engagement) [3, 15, 16]. While results-sharing represents a good research practice, imbued within community/ participant engagement before and during the trial, it is important to maintain those links and relationships with the community and/or individual participants after the completion of the trial [3, 15]. Results-sharing and being responsive to queries related to what eventuated with the data after the completion of research can demonstrate a high accountability, presence beyond the need (an altruistic gesture) and ultimately trust and goodwill for future research [15, 34]. Unfortunately, such post-trial commitments and engagement with participants is underwhelmingly executed and has negative impacts on current research practice [3, 23].

Despite most researchers regarding results dissemination after the study completion to be important, most researchers pointed out a multitude of factors that were deemed to impede the results dissemination to participants and the community [15]. According to previous research, the main barriers to dissemination of research to participants were a lack of early

planning and respective support for the activities from funders, ethical committees and institutional bodies, lack of incentives and training in health communication, different cultural expectations, concerns over potential misunderstanding and misrepresentations of the nature of results, and a general unfamiliarity with the concept and potential methods [17]. Knowledge of barriers is essential to inform and design the appropriate strategies. Barriers identified in this study therefore could be utilized in formative research necessary to develop and build future strategies.

There are clear paucities in the literature around researchers' lack of knowledge on how to share data/results to individual patients or the community [15]. The paucity is also complicated by obvious challenges inherent in cultural, practical, and scientific barriers. For instance, complex (negative) outcomes are difficult to convey to participants or the community where research, health and general literacy is low [17]. Skills to simplify the complexities, time and resources are major needs in such an endeavor and are widely covered in literature related to science communication/translation and public engagement [34, 35]. Despite good intent, there is a risk in conveying scientific information that it may be misinterpreted and thus can be detrimental to the health intervention, increasing unnecessary fear, uncertainty and degrading epistemic trust [36]. Wider literature around community and public engagement emphasizes appropriate scientific translation, and strategies to simplify scientific complexities, such as using arts and social media, which ultimately requires collaboration with a multi-disciplinary team [34, 37].

Other important barriers in results-sharing with the participants or community is the prolonged time between the completion of the trial and the results dissemination; with long time intervals between their participation and outcome of the research, community members may perceive the results to be irrelevant. Malaria, at least in some countries, is sometimes spread through the movement of mobile and migrant populations [38] and reaching out to itinerant populations to share results after a long time has lapsed may not be feasible.

Finally, it is acknowledged that there may be impediments inherent in researchers' career trajectories and institutional policies as well. Conventionally researchers are used to a cycle of securing a grant, conducting research, publishing and disseminating results with policymakers and scientists [21, 22, 39]. The time for planning of research (including the research cycle) is time-constrained and leaves little space for participant or community engagement for results-sharing after the trial [40]. Although transforming in recent years, traditionally, research, academic and funding institutions place much higher emphasis on research outputs (e.g. publications in high impact journals, and policy translations) compared to the engagement with trial participants or community after the trial [41]. Lack of clear instructions and mandates on post-trial results-sharing and little incentive from research institutions and funders also offer flexibility for researchers to overlook the post-trial engagement. There is a clear need to work towards potential methods and strategies on results-sharing with participants or communities after the completion of the trial [3, 15, 16]. Having a policy for data sharing has been proposed as a first step towards the data sharing practice [42]. Discussions/workshops with stakeholders including participants, communities, policymakers, researchers and funders can offer solutions that are both feasible and acceptable to participants (or participant communities) and the wider stakeholders relevant to the local social and research context.

## Researchers' reflections

This study attempted to utilize the formative approach to build evidence incorporating a mixed methods approach. Initially the research was designed with a qualitative component that would be analysed to inform the quantitative survey questions, and that the quantitative

module would be the more illustrative component of the study. After conducting 11 qualitative interviews with malaria researchers, a wealth of insights and opinions were discovered, and have been instrumental in formulating some suggestions for how malaria researchers might approach dissemination at the completion of their trials. The qualitative interview was generally well received and required only a single email invitation. Preliminary findings from qualitative interviews aided authors to tailor and prioritize the thematic areas for further exploration including to inform the questionnaire. The tailoring and designing of qualitative and quantitative methods therefore were complementary to each other and allowed the refining of triangulation of the results presented in this study.

Despite many attempts to encourage participation in the quantitative survey (use of high-quality graphics; click-bait headlines; short scripts of information vs. long scripts; direct links; multiple reminders) the quantitative survey had a very low response rate of 20%. While this is in line with the known challenges of online surveys, it introduces 'non-response bias' leading to a self-selection of only respondents who are engaged in the subject matter and thus skewing the results [43, 44]. It is quite possible that the reason researchers did not participate is for the very reason we are investigating the matter, that it is not a priority, nor common practice for researchers. To mitigate the self-selection bias because of non-response rate in this study, authors shared the findings within their network of researchers and other experts in the field, who validated the results with their practice (and findings). Inherent in interpretive methods utilized in this study, the findings and their interpretations are also influenced by the personal and professional values including experiential knowledge of the authors. In future, interpretive methodology could be broadened to include the opinions of community and stakeholders related to the findings presented in this study. In addition, to respond to the low survey participation incentivized online surveys could be a possible way to enhance the number and diversity of participants.

## Conclusions

Although sharing results and participant engagement after a trial was deemed critical for ethical research and many researchers expressed their intention to do so, it was poorly practiced. Some of the identified barriers could be overcome by early planning and appropriate resourcing, as well as by devising explicit institutional guidance, standard operating procedures and more stringent requirements by funders and ethical review boards to conduct meaningful dissemination. Further research is however needed to inform best practice and methods for post-trial engagement to overcome other identified barriers.

## Supporting information

**S1 Appendix. COREQ guideline.**
(PDF)

**S2 Appendix. SSI guide.**
(PDF)

**S3 Appendix. Survey questionnaire.**
(PDF)

## Acknowledgments

We would like to express our sincere gratitude to 7[th] Clinical Trial Network Meeting members who generously contributed their feedback to validate our findings. We specially thank Naomi Waithira for her time and feedback.

## Author Contributions

**Conceptualization:** Sophie Weston, Kamala Thriemer.

**Formal analysis:** Sophie Weston, Bipin Adhkari, Kamala Thriemer.

**Funding acquisition:** Kamala Thriemer.

**Investigation:** Sophie Weston.

**Methodology:** Kamala Thriemer.

**Project administration:** Sophie Weston.

**Supervision:** Kamala Thriemer.

**Visualization:** Bipin Adhkari.

**Writing – original draft:** Sophie Weston, Bipin Adhkari.

**Writing – review & editing:** Sophie Weston, Bipin Adhkari, Kamala Thriemer.

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
