## [Decision Letter · Decision Letter 0]

16 May 2023

PGPH-D-23-00256

Sharing results with participants (and community) in malaria related research: perspectives and experience from researchers

Dear Dr. Thriemer,

Thank you for submitting your manuscript to PLOS Global Public Health. After careful consideration, we feel that it has merit but does not fully meet PLOS Global Public Health’s publication criteria as it currently stands. Therefore, we invite you to submit a revised version of the manuscript that addresses the points raised during the review process.

Please carefully address several comments by both reviewers and please conduct careful proofreading of the manuscript and editing for language revisions to improve readability and address any grammatical/language errors, in addition to the technical comments by the reviewers. 

We look forward to receiving your revised manuscript.

Kind regards,

Prashanth Nuggehalli Srinivas, MBBS, MPH, PhD

Academic Editor

Journal Requirements:

2. Please provide separate figure files in .tif or .eps format.

Additional Editor Comments (if provided):

Reviewers' comments:

Reviewer's Responses to Questions

**Comments to the Author**

1. Does this manuscript meet PLOS Global Public Health’s publication criteria? Is the manuscript technically sound, and do the data support the conclusions? The manuscript must describe methodologically and ethically rigorous research with conclusions that are appropriately drawn based on the data presented.

Reviewer #1: Partly

Reviewer #2: Yes

2. Has the statistical analysis been performed appropriately and rigorously?

Reviewer #1: I don't know

Reviewer #2: N/A

3. Have the authors made all data underlying the findings in their manuscript fully available (please refer to the Data Availability Statement at the start of the manuscript PDF file)?

Reviewer #1: Yes

Reviewer #2: No

4. Is the manuscript presented in an intelligible fashion and written in standard English?

Reviewer #1: Yes

Reviewer #2: Yes

5. Review Comments to the Author

Reviewer #1: Dear Editor,

RE: Review Report for PGPH-D-23-00256: Sharing results with participants (and community) in malaria related research: perspectives and experience from researchers

Thank you for the opportunity to review this interesting manuscript. I find the topic being addressed (return of trial results) quite topical and important. It is also interesting that the authors attempt to place this topic in the broad context of community engagement in research. The paper is fairly well written, although there are a number of minor issues that the authors need to address and or clarify.

1. The manuscripts needs to be proofread to ensure accuracy of language.

2. Introduction

a) The title of the manuscript suggests that the concern of the paper is about sharing results from trial in Malaria studies. However, immediately after citation number 4, in a spur-of-the-moment, the manuscript seems to digress to ‘community engagement’. The citations that follow focus on broad community engagement. This seeming digression can perhaps be qualified by indicating that the return of trial results is a critical component in CE, then go ahead to speak briefly about community engagement.

b) In the sentence which begins on p.4 and ends on p.5, there is a phrase, “much less is known about whether and how dissemination of results is done …”. Instead of “whether” it should be ‘the extent to which’. Asking ‘whether’ is so limiting because with this question, you simply need at least only one instance in which the results were shared to get an affirmed answer, and yet I think we should be mainly interested in how often trial results are shared.

c) Study aim and objectives: The stated aim in the abstract is somewhat different from that in the rest of the work beginning from the introduction. For example, in the abstract the authors talk of “… methods and strategies” while in the introduction they talk of “… methods or strategies”. Again, “Perspectives” are indicated in the abstract but not in the introduction, while “attitudes” are indicated in the introduction but not in the abstract. (The listing of the results themes on p.6 Par. 1 suggests that the authors’ objectives are around “general opinion” (attitudes?), “Practices” and “Perspectives”).

d) Further, in the introduction the authors have what they call “a secondary aim”. Why are they downplaying this aim and yet there are substantial and significant results for it in the manuscript? In addition, while stating this “secondary aim” the authors talk of “… experiences or barriers …”. Does the use of a disjunction ‘or’ between these words imply that they are synonyms or alternatives? Generally, there is a need to be consistent and clear, including conceptually, about the aims and objectives of the manuscript/study. The main concepts of which the distinctions are not clear in this work are: Attitudes; perceptions, perspectives, and general opinion. For example, is ‘general opinion’ …” conceptually identical with ‘attitude’, or ‘perspective’?

3. Methods

1) On p. 8, par. 2, while describing webinar attendees, the authors indicate that attendees “represented South Asia, Southeast Asia …”. Is the concept of ‘representation’ being used in its strict sense or to mean that the participants simply ‘came from’ these various regions or even one or two countries from each of these regions? If the authors are sure about the representativeness of these attendees, they should provide further information to qualify this claim of representation.

2) Still on 8, par. 2, the sample size (11 interviews leading to data saturation) seems to have been determined after data analysis as opposed to the usual norm of determining such saturation during interviews. This casts doubt on the credibility of their claim of data saturation.

4. Results:

a) On p. 8, par. 3, for the qualitative phase of the study, the investigators/authors mailed (invited) only 13 potential participants even though we know that saturation can sometimes be achieved at 25 interviews, or even more. Why did they target only 13, and what was the approximate number of the target study population?

b) For the quantitative (survey) phase of this study, the sampling frame was 293 potential participants and ultimately only 44 were able to participate. Unfortunately, the authors do not indicate what the actual target sample size was, how it was determined and hence, whether a sample of 44 sufficiently powered the results.

c) On p. 9, under rationale, par 2, the authors state that “Respondents of the SSIs agreed in principle that results ….”. It is important to use specific quantifiers to give an idea about the extent to which this agreement was shared among the respondents, e.g. most, majority, a few, an almost equal number etc. ‘Some’ can be used but cautiously because apart from indicating ‘at least one or all) it does not give a good picture of how widespread the opinion is.

d) Some sections of the findings such as on pp. 12 and 13 verbatim quotes from participants are introduced by verbatim quotes of the questions posed by the interviewer, but this is not the case with other quotes. There should consistency in the style of introducing the quotes, preferably, paraphrasing questions into descriptive statements.

e) In the last line on p.13, the words “participants only” seem to be intended as a noun. If that’s true, then it should be one hyphenated work ‘participants-only’.

f) On p. 14, the second paragraph begins as “This was similarly reported…”. What is “this” referring to? Is it a continuation of the previous paragraph? In the same sentence, in reporting the results in the form proportions or fractions, the denominator is 35, but elsewhere it is 42. Elsewhere, the authors indicate 14/33; 3/33 on p.17, 13/25 on p. 18, 9/40 on p.19, etc. Why? Is the denominator representing the number of people who responded to particular questions or something else? Clarity is needed.

g) Table 2, pp 15 to 16: What is the rationale of presenting this information in a table form? Secondly, the authors indicate that this (text in the table) is an “author-edited narrative”. This gives an impression that these were paraphrased views of participants. If they are not paraphrased, then they should be written like any other verbatim quotes and the edits they mention can be indicated with appropriate punctuation such as square brackets to indicate that some text was added or skipped, and sic where applicable.

h) On p. 17, in the middle of the second paragraph, there is a phrase: ..”…time lag was mentioned by the majority (14/33, 42.2%) … These figures are not a majority.

i) On p. 18, the claim in the sentence beginning the paragraph immediately the first direct quote is factually contentious. It is not true, as far I know, that rural populations in Africa are generally more mobile than those in urban settings. Instead the reverse is generally true. The authors seem to be having mind certain specific populations (exceptions) that fit these attributes (rural and highly mobile), but it would be wrong to generalize at a continental or regional (Sub-Saharan Africa) level. They need to qualify their claim or specify that these were the attributes of the specific populations where the Malaria trials they refer to took place.

j) Ethical considerations are not described.

5. Discussion

a) P. 20, second sentence (third line) “… deemed to have rights to know …”. The concept of rights can be controversial. It can be replaced with ‘deserve to know’ except if the concept of ‘rights’ was explicitly introduced by the respondents. If this statement on rights is intended to be a normative one as opposed to descriptive, it is fine but it needs to be clear.

b) P. 20, second par, the first statement is clearly reporting the findings of this study but at its end there is a citation! This is common throughout the discussion section. What does this mean?

c) P.21, par 2, last sentence: I find the concept of ‘epistemic trust’ unfamiliar. Generally there is a growing tendency to misuse use the concept of ‘epistemology’ (such as the growing rhetoric about ‘epistemic justice/injustice’). Since it is clear that what the authors are referring to is the increasing ‘doubt about the credibility of research results’ they can choose to replace the phrase ‘declining epistemic trust’ with a different phrase such as ‘increasing allegations of lack of scientific integrity, or credibility of research data/findings’ to avoid potential conceptual controversies.

d) P. 23, last par. The paragraph is introduced as limitations of the study, although it is not clear how the content of this particular paragraph can be regarded as limitations.

e) Generally, the discussion could have been better. A lot of space is dedicated to paraphrasing the results of the study and showing how they are corroborated by results from other similar studies. Whereas this is part of what needs to be done in the discussion section, there is need to directly draw inferences and or implications from the results – something like what has been done on p. 24, second paragraph. In this paragraph, the authors have done quite well in attempting to draw the implication of very low enthusiasm on the part of researchers to respond to their request to participate in the study – that is, that it could be that researchers do not find the topic of sharing trial results to be of priority. (Disclaimer about this comment: This type (not necessarily quality) of discussion is common in many empirical Social Science papers, and yet my background is philosophy where there is a lot of emphasis on independent examination of the reported views, and drawing implications and or further inferences from them).

Sincerely,

John Barugahare, Ph.D.

Senior Lecturer

Department of Philosophy, Makerere University

Reviewer #2: Thank you for the opportunity to review the manuscript associated with your research. I have attached my comments in a separate document. The comments relate to the requirement for all data underlying the findings to be made fully available, as well as issues specifically relating to the paper.

6. PLOS authors have the option to publish the peer review history of their article (what does this mean?). If published, this will include your full peer review and any attached files.

**Do you want your identity to be public for this peer review?** For information about this choice, including consent withdrawal, please see our Privacy Policy.

Reviewer #1: **Yes: **John Barugahare

Reviewer #2: No

---

## [Decision Letter · Decision Letter 1]

25 Jul 2023

PGPH-D-23-00256R1

Sharing results with participants (and community) in malaria related research: perspectives and experience from researchers

Dear Dr. Thriemer,

Thank you for submitting your manuscript to PLOS Global Public Health. After careful consideration, we feel that it has merit but does not fully meet PLOS Global Public Health’s publication criteria as it currently stands. Therefore, we invite you to submit a revised version of the manuscript that addresses the points raised during the review process.

We look forward to receiving your revised manuscript.

Kind regards,

Prashanth Nuggehalli Srinivas, MBBS, MPH, PhD

Academic Editor

Journal Requirements:

Additional Editor Comments (if provided):

Please make the minor revisions suggested by R1. In addition, please revisit the comments of R2 on the previous version of this manuscript, especially his comments related to Discussion section as mentioned by the reviewer on this version.

Reviewers' comments:

Reviewer's Responses to Questions

**Comments to the Author**

1. If the authors have adequately addressed your comments raised in a previous round of review and you feel that this manuscript is now acceptable for publication, you may indicate that here to bypass the “Comments to the Author” section, enter your conflict of interest statement in the “Confidential to Editor” section, and submit your "Accept" recommendation.

Reviewer #2: All comments have been addressed

2. Does this manuscript meet PLOS Global Public Health’s publication criteria? Is the manuscript technically sound, and do the data support the conclusions? The manuscript must describe methodologically and ethically rigorous research with conclusions that are appropriately drawn based on the data presented.

Reviewer #2: Yes

3. Has the statistical analysis been performed appropriately and rigorously?

Reviewer #2: Yes

4. Have the authors made all data underlying the findings in their manuscript fully available (please refer to the Data Availability Statement at the start of the manuscript PDF file)?

Reviewer #2: No

5. Is the manuscript presented in an intelligible fashion and written in standard English?

Reviewer #2: Yes

6. Review Comments to the Author

Reviewer #2: The paper is much improved even though the researchers did not seem to fully understand the suggestion made by Reviewer 2 regarding the Discussion.

The researchers should consider including a heading at line 547. For example, it could read 'Researchers’ Reflections'. In addition, a heading, such as 'Conclusion', should be added at line 570. The addition of these headings would facilitate reading the manuscript.

7. PLOS authors have the option to publish the peer review history of their article (what does this mean?). If published, this will include your full peer review and any attached files.

**Do you want your identity to be public for this peer review?** For information about this choice, including consent withdrawal, please see our Privacy Policy.

Reviewer #2: No

---

## [Editor Report · Decision Letter 2]

8 Aug 2023

Sharing results with participants (and community) in malaria related research: perspectives and experience from researchers

PGPH-D-23-00256R2

Dear Dr. Thriemer,

We are pleased to inform you that your manuscript 'Sharing results with participants (and community) in malaria related research: perspectives and experience from researchers' has been provisionally accepted for publication in PLOS Global Public Health.

Best regards,

Prashanth Nuggehalli Srinivas, MBBS, MPH, PhD

Academic Editor